# Dynamic Crushing Behavior of Ethylene Vinyl Acetate Copolymer Foam Based on Energy Method

**DOI:** 10.3390/polym15143016

**Published:** 2023-07-12

**Authors:** Yueqing Xing, Xiya Guo, Guowei Shu, Xiaolong He

**Affiliations:** College of Bioresources Chemical and Materials Engineering (College of Flexible Electronics), Shaanxi University of Science & Technology, Xi’an 710021, China

**Keywords:** ethylene vinyl acetate copolymer (EVA), density, thickness, dynamic crushing, *SEA*, impact energy-to-thickness ratio

## Abstract

This paper aimed to experimentally clarify the dynamic crushing mechanism and performance of ethylene vinyl acetate copolymer (EVA) and analyze the influence of density and thickness on its mechanical behavior and energy absorption properties under dynamic impact loadings. Hence, a series of dynamic compression tests were carried out on EVA foams with different densities and thicknesses. When the impact energy is 66.64 J, for foam with a density of 150 kg/m^3^, the maximum contact force, maximum displacement, maximum strain, absorbed energy, and specific energy absorption (*SEA*) increased by 20 ± 2%, −38.5 ± 2%, −38.5 ± 2%, 4 ± 2%, and 105 ± 2%, respectively, compared to foam with a density of 70 kg/m^3^. The ratios of absorbed energy to impact energy for different thickness specimens are almost equal. The specimen density has no effect on the efficiency of energy absorption and has a greater effect on the *SEA*. Meanwhile, when the impact energy-to-thickness ratio is 1680 J/m, compared to foam with a thickness of 30 mm, the maximum contact force, maximum displacement, maximum strain, absorbed energy, and *SEA* for foam with a thickness of 60 mm increased by 28.5 ± 2%, 211.3 ± 2%, 56.6 ± 2%, 100.8 ± 2%, and 0.4 ± 0.5%, respectively. When the impact energy is 66.64 J, compared to foam with a thickness of 30 mm, the maximum contact force, maximum displacement, maximum stain, absorbed energy, and *SEA* for foam with a thickness of 60 mm increased by −42.5 ± 2%, 163.5 ± 2%, 31.7 ± 2%, 4.1 ± 2%, and 4.1 ± 2%, respectively. The *SEA* of two different-thickness EVA specimens is almost equal, about 2.8 J/g. The ratios of absorbed energy to impact energy for different thickness specimens are almost equal, both at 72%. The specimen thickness has no effect on the efficiency of energy absorption and has a greater effect on the maximum contact force. In the range of impact energy, thickness, and density studied, the absorbed energy and *SEA* are not affected by the thickness of EVA specimens and are determined by the impact energy. The density has no significant effect on the absorbed energy but has a greater effect on the *SEA*. However, for EVA foams, the greater the density, the greater the mass, and the higher the cost. Taking into account lightweight and cost factors, when optimizing cushioning design within a safe range, we can choose EVA foams with a smaller density and thickness.

## 1. Introduction

EVA foaming material is a non-toxic foaming material with excellent foaming properties that is widely used in sports equipment, sports shoes, seat cushions, and other industries. EVA foam material is widely used because of its light weight, comfort, good plasticity, excellent elasticity, and other properties [1,2].

At present, the research focus on EVA foam is mainly on its production and preparation technology [3,4]. For lightweight design, EVA foams have the considerable potential to further increase energy absorption, which has drawn increasing interest recently [5]. 

Many scholars have studied the dynamic buffering energy absorption characteristics of foam materials. Wang conducted dynamic buffering tests based on the principles of free drop and energy absorption to analyze the influence of honeycomb structure factors on impact performance [6]. Zhou studied the mechanical properties of aluminum honeycomb under axial direction dynamic impact loads [7]. Cronin and Ouellet studied the effect of strain rate on the mechanical properties of low-density polyethylene, polystyrene, and polypropylene [8]. Tan et al. proposed a new method to determine the density point in the stress–strain curve of foam [9]. Petrone studied the behavior of short flax fibers and continuous flax fibers reinforced with polymer cores under low-impact loads [10]. Tan and Akil took fiber metal laminate as the surface, formed a sandwich structure with a polypropylene honeycomb core, and conducted a series of low-speed impact tests [11]. Wang introduced the velocity sensitivity of aluminum honeycomb under high-speed axial impact and proved that the increase in impact velocity promotes energy absorption [12]. Sanchez-Saez studied the dynamic impact behavior of agglomerated cork, analyzing the influence of the specimen thickness on the energy absorption capacity, contact force, displacement, and strain. It was observed that the energy absorbed by agglomerated cork did not depend on the specimen thickness but only on the impact energy [13]. Zhang Jingjing also studied the influence of thickness, gas content, and thickness span ratio on the impact performance of hexagonal honeycomb cardboard on an out-of-plane surface through dynamic impact tests. The ability of honeycomb boards to absorb energy is independent of their thickness and depends on their impact energy [14]. Bouix studied the effects of density and strain rate on the energy absorption efficiency of foamed polypropylene in the range of engineering strain rate 0.01~1500 s^−1^ [15]. Some studies have analyzed metal tubular structures filled with agglomerated cork subjected to low-velocity impact [16,17] and high-velocity impact [18]. Lee studied the effect of the density of polypropylene foams on impact energy absorption and stress–strain curves [19]. The mechanical behavior under quasi-static compression of the agglomerated cork is typical of a cellular material. Under static compressive loads, agglomerated cork shows a linear elastic behavior due to the bending resistance of the cell walls. Mechanical and energy absorption properties are strongly influenced by density [20,21,22].

However, there is a paucity of information about the mechanical behavior of EVA foam subjected to dynamic loads based on the energy method available in the literature. Yueqing Xing [23] studied the static crushing responses of expanded polypropylene (EPP) foam and found that EPP foam density has a significantly greater influence on static compressive performance than foam thickness. Thus, the aim of this work is to analyze the dynamic crushing behavior of the EVA foam and the influence of material thickness and density on the energy absorption capacity of the foam structure based on the previous paper. Therefore, to optimize the design of structures with EVA foams, it is necessary to study the behavior of this material under dynamic impact conditions. Dynamic impacting tests were performed on specimens of four different thicknesses and four different densities, analyzing the maximum contact force, maximum displacement and strain, absorbed energy, and *SEA*.

In the process of the dynamic impact test of the EVA foams, three schemes were made by changing the heavy hammer weight and the impact falling height. The first scheme was to study the density and thickness of specimens’ effects on the mechanic’s performance and the energy absorption characteristics under certain impact energy conditions. The second scheme was to study specimens with different thicknesses under certain impact energy-to-thickness ratios. The third scheme was under different impact energies when the density and thickness of the specimens were constant. The acceleration–time curves of EVA foams with different thicknesses and densities were obtained by means of dynamic impact tests and related standards. Then, by changing the impact height to obtain different impact energy and impact energy-to-thickness ratios, the evaluating indicators of contact force, maximum contact force, maximum displacement, maximum strain, energy absorption, and *SEA* were obtained for the EVA foams.

## 2. Materials and Methods

### 2.1. Materials

The EVA foam hereby acquired was from Suzhou Shunsheng Packaging Material Co., Ltd., Suzhou, China, with a hardness of 38 HA, 45 HA, 55 HA, 60 HA, and 70 HA and a corresponding density of 70 kg/m^3^, 85 kg/m^3^, 110 kg/m^3^, 150 kg/m^3^, and 175 kg/m^3^, respectively. Specimens had a cross-section of 150 mm × 150 mm and four different thicknesses (30 mm, 40 mm, 50 mm, and 60 mm). Specimen labels include a letter and two numbers: for example, S represents the uniform code of the test specimen, the following two-digit number is the hardness, and the last two-digit number is the thickness. Thus, specimen S60-30 is the specimen with a hardness of 60 HA and a thickness of 30 mm.

### 2.2. Impact Test Equipment and Method

The test equipment is an XG-HC impact testing machine, made by Xi’an Guangbo Testing Equipment Co., Ltd., Xi’an, China. The experimental method was in strict accordance with GB/T 8167-2008 [24]. The impact test specimens were prepared for more than 24 h in an environment of 27 °C and 72% relative humidity. In all tests, the specimens were placed centered on the center point of the lower pressure plate of the testing machine. The maximum drop height of the XG-HC testing machine is 1200 mm, and the minimum mass and maximum mass of the impact sliding table are 7 kg and 50 kg, respectively. Meanwhile, according to GB/T 8167-2008, the thickness of the test specimen should not be less than 25 mm. 

The dynamic impact test system is composed of an XG-HC impact testing machine, an acceleration sensor, a displacement sensor, and a data acquisition and processing system. The data acquisition and processing system includes a charge amplifier, the description of various curves, and the storage and management of various data files. Figure 1a,b shows the EVA specimens and impact equipment.

For characterization of pore morphology, the samples were quenched in liquid nitrogen, sprayed with gold, and observed using a JSM6390LV scanning electron microscope produced by Japan JEOL Corporation. Figure 1c,d shows the scanning electron microscope (SEM) structures of EVA specimens with hardness of 45 HA and 70 HA, respectively.

### 2.3. Impact Compression Test Equipment

Figure 1 shows the dynamic impact process of a heavy hammer on the EVA foam. The specimen was placed in the middle of the rigid support platform of the testing machine. The heavy hammer is fixed to the sliding table through the fixing device. The weight of the heavy hammer and the dropping height of the sliding table can be set and changed according to the requirements of the test. The heavy hammer and the sliding table could be dropped freely along the smooth guide column of the testing machine, and the dropping height *h* could be set according to the test requirements. The coordinate system is shown in Figure 2.

The acceleration sensor was installed in the middle of the sliding table. When the heavy hammer and the sliding table impact the EVA foam, the acceleration sensor outputs a voltage signal through the charge amplifier. After LPF (low-pass filtering) and an A/D converter, the impact acceleration–time curve of the specimen was generated after analysis and processing by the data analysis and processing software. The XG-HC machine and the principles of its measurement and control system adopted in this experiment are shown in Figure 3.

### 2.4. Impact Characteristic Criteria

Contact force is a relevant parameter in the study of energy absorption elements [25]. In a dynamic compression test, the absorbed energy *E*(*t*) of the EVA foam could be expressed as: (1)$$E\left(t\right)=\int_{0}^{x}F\left(t\right)dx$$
(2)$$F\left(t\right)=ma\left(t\right)$$
where *E*(*t*) is the absorbed energy by the EVA foam. *F*(*t*) is the contact force, *m* is the weight of the heavy hammer and the slide table (kg), *x* is the contact force displacement, and *a* is the instantaneous acceleration (m/s^2^).

The absorbed energy is related to the contact force *F*(*t*) and the displacement *x*, so the contact force is the relevant parameter for studying the absorbed energy of foamed materials. For each test, the contact force could be calculated as a function of the contact time.

According to GB/T8167-2008, the acceleration–time curve of each impact can be obtained, which is the half-sine wave as shown in Figure 4 and can be expressed as:(3)$$G=G_{m}\sin\frac{\pi }{\tau }\left(t-t_{1}\right)$$
where *G* is the acceleration (m/s^2^), *G*_m_ is the maximum acceleration (m/s^2^), and *τ* is the difference value between time *t*_2_ and *t*_1_. In the dynamic impact test, the instantaneous impact force generated by the initial contact between the impact heavy hammer and sliding table on the specimen is defined as the contact force and can be obtained by Equations (2) and (3):(4)$$F\left(t\right)=ma\left(t\right)=mG_{m}\sin\frac{\pi }{\tau }\left(t-t_{1}\right)$$

It can be seen that the magnitude of the contact force is related to the mass of the heavy impact hammer and sliding table and the instantaneous acceleration. Then the impact stress *σ*(*t*) on the specimen is expressed as:(5)$$\sigma \left(t\right)=F\left(t\right)/A=ma\left(t\right)/A$$
where *A* is the specimen area.

In the process of the heavy hammer and sliding table impacting the specimen, the transient velocity is expressed as:(6)$$v\left(t\right)=v_{i}-\int_{0}^{t}a\left(t\right)dt$$

Displacement (impacting compressive deformation) *x*(*t*) changes over time as follows (Figure 1 shows that the origin of displacement coordinate is the upper surface without impact, and downward is positive):(7)$$x\left(t\right)=\int_{0}^{t}v\left(t\right)dt=\int_{0}^{t}\left[v_{i}-\int_{0}^{t}a\left(t\right)dt\right]dt$$

The energy absorbed *E*(*t*) by the specimen in the dynamic compression process is expressed as:(8)$$E\left(t\right)=F\left(t\right)\cdot x\left(t\right)=\int_{0}^{t}F\left(t\right)\left[v_{i}-\int_{0}^{t}a\left(t\right)dt\right]dt$$

It is assumed that the deformation along the thickness direction is uniform when the porous EVA material is subjected to uniaxial impact compression. The strain *ε*(*t*) can be obtained from Equation (7) as follows:(9)$$\varepsilon \left(t\right)=x\left(t\right)/T=\left[\int_{0}^{t}\left[v_{i}-\int_{0}^{t}a\left(t\right)dt\right]dt\right]/T$$
where *T* is the specimen thickness. If the loss of some energy, such as friction energy and sound energy, between the weight and air during the impact process of falling impact is not taken into account, the energy conversion relationship during the impact load exerted by the weight of the heavy hammer and sliding table on the specimen under ideal conditions can be described as follows:(10)$$mgh=mv_{i}^{2}/2=AT\int_{0}^{\varepsilon_{max}}\sigma d\varepsilon$$
where *h* is the falling height and *v_i_* is the impact velocity. The energy released by the rebound of the specimen after impact compression is expressed as:(11)$$\mathrm{mg}h_{f}=mv_{f}^{2}/2=AT\int_{\varepsilon_{f}}^{\varepsilon_{max}}\sigma_{x}d\varepsilon$$
where *h_f_* is the heavy hammer rebound height, *v_f_* is the velocity of the heavy hammer bouncing, and *ε_f_* is the strain of the specimen after removal of the compression load. Then the energy absorbed *E_loss_* by the specimen during the impact can be described as:(12)$$E_{loss}=mg\left(h-h_{f}\right)=m\left[\left(v_{i}^{2}-v_{f}^{2}\right)/2\right]=AT\left[\int_{0}^{\varepsilon_{f}}\sigma d\varepsilon +\int_{\varepsilon_{f}}^{\varepsilon_{max}}\left(\sigma -\sigma_{x}\right)d\varepsilon \right]$$

Because the acceleration sensor has been installed in the middle of the sliding table, the measured impact acceleration is the actual impact acceleration, which has removed the mechanical energy loss.

*SEA* is defined as the energy absorbed by the specimen per mass, which is an important parameter to measure the energy absorption of the EVA foam. The larger the *SEA*, the better the energy absorption effect of the EVA foam. Specific absorption energy is defined as follows:(13)$$SEA=\frac{E_{loss}}{m}$$
where *m* is the mass of the specimen.

## 3. Results and Discussion

In order to evaluate the influence of specimen thickness on the dynamic mechanical properties of the EVA foam, the concept of impact energy-to-thickness ratio was defined [13]. The method is based on the change in impact energy and the impact energy-to-thickness ratio. It can obtain the maximum contact force, maximum displacement, maximum strain, and energy absorption. These evaluation indexes can be used to analyze the effects of density and thickness on the dynamic properties and energy absorption characteristics of the EVA foams during the dynamic impact process. 

The theoretical impact energy can be calculated according to Equation (10). Several levels of impact energy were selected for each thickness to obtain the same impact energy-to-thickness ratio. About 108 specimens are tested. In Table 1, the levels of impact energy, the impact energy-to-thickness ratios, and the number of specimens tested are shown to analyze the influence of thickness on the dynamic impact performance of EVA foams.

Several levels of impact energy were selected for each density to analyze the influence of density on the dynamic impact performance of EVA. About 108 specimens are tested. In Table 2, the levels of impact energy and the number of specimens tested are shown.

### 3.1. The Contact Force

The contact force in the impact process refers to the force between the heavy hammer and the specimen during the impact and the process by which the specimen rebounds to the highest point at which the heavy hammer and the specimen separate. During the contact between the heavy hammer and the specimen, the velocity, kinetic energy, and specimen shape variables will change, along with the processes of energy conversion and dissipation. 

For each test, the contact force as a function of the contact time is recorded. It can be obtained from Equation (3). As an example, the contact force curves of specimens with different thicknesses for impact energies around 64.64 J are shown in Figure 5a. All curves exhibit some oscillation due to the vibration of the impacting testing device and specimens, as mentioned regarding other foamed materials [15].

It can be seen from Figure 5a that with the increase in specimen thickness, the maximum contact force decreases gradually, and the contact time increases. The change in maximum contact force with thickness may be affected by the damping of the material. The greater the thickness of the specimen, the greater the mass, which will lead to greater damping and reduce the acceleration of the impactor and thus the maximum contact force [25]. As shown in Figure 5a, when EVA foam thickness increases from 30 mm to 60 mm, the maximum contact force decreases by −42.8% (5.47–3.15 KN) and the contact time increases by 59.9% (9.87–15.79 ms). 

When the impact energy-to-thickness ratio is 1680 J/m, the contact force and time curves are shown in Figure 5b. The different damping of specimens caused by different thicknesses leads to a change in contact force and contact time with thickness. The maximum contact force and the contact time gradually increase with the increase in specimen thickness. This is because when the impact energy-to-thickness ratio is constant, the thicker the specimen is, the greater the impact energy, and the greater the maximum contact force and contact time. As shown in Figure 5b, when EVA foam thickness increases from 30 mm to 60 mm, the maximum contact force increases by 35.4% (4.91–6.87 KN), and the contact time increases by 54.3% (10.30–15.89 ms).

As can be seen from Figure 5c, when the impact energy is 66.64 J and the specimen thickness is constant, as the density of the EVA specimen increases, the maximum contact force is greater and the contact time is shorter. According to Formula 1, under the condition that the thickness and impact energy of the specimen are constant, the larger the density of the specimen, the smaller the shape variable, and the larger the contact force will be. As shown in Figure 5c, when EVA foam density increases from 70 kg/m^3^ to 175 kg/m^3^, the maximum contact force increases by 32.1% (3.94–5.21 KN) and the contact time decreases by −23.9% (13.89–10.57 ms). As shown in Figure 5d, dynamic impact tests were carried out on specimen S38-50 with different impact energies. With the increase in impact energy, the maximum contact force is larger, but the contact time is similar, and the maximum contact force appears at the same time. As shown in Figure 5d, when impact energy increased from 50.18 J to 82.32 J, the maximum contact force increased by 77% (3.82–6.76 KN).

### 3.2. The Maximum Contact Forces, Displacement and Strain

Maximum contact force can be obtained by Equation (4). Maximum displacement and maximum strain can be obtained from Equations (6), (7), and (9). Maximum contact force, maximum displacement, and maximum strain of specimens with different thicknesses under different impact energy-to-thickness ratios are shown in Table 3. The maximum contact force, maximum displacement, and maximum strain of specimens with different densities under different impact energies are shown in Table 4. Figure 6a shows the maximum contact force vs. impact energy-to-thickness ratio for the different thickness specimens. The maximum contact force is directly proportional to the impact energy-to-thickness ratio. The maximum contact force increased with the increment of the impact energy-to-thickness ratio for all the different thickness specimens. Although this result presents a large dispersion, for the same impact energy-to-thickness ratio, the value of the maximum contact force is similar. 

Figure 6b shows the maximum contact force vs. impact energy for the different density specimens. The maximum contact force increased with the increment of the impact energy for all the different density specimens. Under the same impact energy, the greater the density of the specimen, the greater the maximum contact force, and the smaller the density of the specimen, the smaller the maximum contact force.

During the impact test, if the influence of damping is ignored, the factor affecting the impact displacement is related to the change in velocity, and the change in velocity is only related to the drop height of the impactor. Therefore, the factor affecting the impact displacement is the drop height of the impactor [14]. The curves of maximum displacement and impact energy-to-thickness ratio are shown in Figure 7a. Figure 7b shows the variation of the strain with the impact energy-to-thickness ratio. As shown in Figure 6a and Figure 7a,b, when the impact energy-to-thickness ratio is 1680 J/m, the specimen thickness increases from 30 mm to 60 mm, the maximum contact force, maximum displacement, and maximum strain increase by 39.9%, 211.3%, and 55.7% (4.91–6.87 KN, 10.93–34.01 mm, and 0.36–0.57), respectively.

The maximum displacement–impact energy and maximum strain–impact energy curves for specimens with different densities are shown in Figure 7c,d. For specimens with any density, the maximum displacement and maximum strain increase when the impact energy increases. For the same impact energy, the higher the specimen density, the smaller the displacement and strain. As shown in Figure 6b and Figure 7c,d, when the impact energy is 66.64 J, the specimen density increases from 70 kg/m^3^ to 150 kg/m^3^, the maximum contact force, maximum displacement, and maximum strain increase by 20%, −38.4%, and −38.4% (3.94–4.73 KN, 33.16–20.41 mm, and 0.66–0.41), respectively. The influence that impact energy has on the maximum displacement and maximum strain of EVA materials with different densities and thicknesses is in accordance with other foam materials [27,28].

### 3.3. Absorbed Energy

The energy at each instant in time was calculated by a double integration process from the contact force–time curve [29]. The cushioning material mainly absorbs the impact energy through the plastic deformation of the material during the impact process so as to reduce the damage caused by the impact force, so the cushioning material is required to absorb the impact energy as much as possible. The main indexes for evaluating energy absorption are total absorbed energy and *SEA*. The absorbed energy can be obtained by Equation (12), and the *SEA* can be obtained by Equation (13). In the dynamic compression test, the ideal state of the impactor impacting the specimen is to convert all the impact energy into absorbed energy, but in fact, due to friction loss, the absorbed energy is less than the impact energy.

When the specimen thickness changes and the impact energy-to-thickness ratio is constant, and when the specimen thickness remains the same and the impact energy/thickness ratio changes, the absorbed energy and specific absorbed energy data of four kinds of EVA materials with different thicknesses are shown in Table 5. When the specimen density changes and the impact energy is constant, and when the specimen density remains the same and the impact energy changes, the absorbed energy and specific absorbed energy data of EVA materials with different densities are shown in Table 6. The effect of thickness on the absorbed energy was analyzed, taking into account the influence of the impact energy/thickness ratio. Figure 8a shows the relationship between the absorbed energy and the impact energy-to-thickness ratio for the four specimen thicknesses studied. Figure 8b shows the relationship between the absorbed energy versus impact energies for the EVA specimens with different thicknesses. The absorbed energy is related to the displacement of other cellular materials [30]. The displacement increases when the impact energy-to-thickness ratio is increased (Figure 7a), as does the absorbed energy. As shown in Figure 8a, when the impact energy-to-thickness ratio is 1680 J/m, the specimen thickness increases from 30 mm to 60 mm, the absorbed energy, *SEA,* increases by 100.8% and 0.4% (36.11–72.52 J, 2.81–2.82 J/g), respectively. The *SEA* of the two-thickness EVA foams is almost the same, indicating that the specimen thickness has no effect on the energy absorption capacity; this behavior differs from that of polymeric foam [25].

As shown in Figure 8b, when the impact energy is 66.64 J, the specimen thickness increases from 30 mm to 60 mm, the absorbed energy, *SEA*, and the ratio of absorbed energy to impact energy increase by 4.1%, 0.4%, and 4% (47.77–49.74 J, 2.81–2.82 J/g, and 71.7–74.6%), respectively. Figure 8a,b shows that the absorbed energy of specimens with different thicknesses is determined by impact energy and is less affected by specimen thickness. In Figure 8b, the relationship between the impact energy and the absorbed energy for all specimens shows a good correlation with a straight line (R^2^ equal to 0.987). Therefore, within the range of impact energy and thickness studied, the specimen thickness has no significant influence on the absorbed energy, just like EPP foams [29]. Figure 8c shows the relationship between *SEA* and the impact energy for the four specimen thicknesses studied. When the thickness of an EVA specimen is constant, the higher the impact energy, the higher the *SEA*. It can be seen from Figure 8c that the energy absorption of an EVA specimen with any thickness is basically the same under a certain impact energy, while an increase in thickness will increase the mass of the EVA specimen and decrease *SEA*. Therefore, when the impact energy is constant, the larger the specimen thickness, the smaller the *SEA* is.

Figure 8d shows the relationship between the absorbed energy and the impact energy for the four EVA specimen densities studied. When the density is fixed, the impact energy increases, and the absorption energy also increases. Therefore, for a specimen of any density, the absorbed energy increases with the increase in impact energy, and the relationship between absorbed energy and impact energy is linear. When the impact energy is fixed, the absorbed energy of a specimen with any density is basically the same. At the same time, it can be clearly seen that the data in Figure 8d have a good correlation with the straight line.

The curves of *SEA* versus impact energy for the EVA specimens with four different densities are shown in Figure 8e. It can be seen that for specimens of any density, the greater the impact energy, the greater the specific absorption energy. For the same impact energy, the larger the specimen density, the smaller the *SEA*. As shown in Figure 8d,e, when the impact energy is 66.64 J, the specimen density increases from 70 kg/m^3^ to 150 kg/m^3^, the absorbed energy, *SEA,* and the ratio of absorbed energy to impact energy increase by 4.1%, −51.4%, and 4% (47.77–49.74 J, 1.36–0.66 J/g, and 71.7–74.6%), respectively. The difference in *SEA* between the two density specimens shows that the density has a great influence on the energy absorption capacity.

The EVA foam has good resilience after dynamic impact. By comparing the compression size of specimen S45-30 before and after dynamic impact, when the impact energy is 66.64 J, the thickness of the specimen before impact is 30 mm, and the thickness after impact is 28.86 mm. This shows that the EVA foam has good resilience.

Due to the increasing attention and concern for lightweight design in the packaging industry, it is required that the energy absorption material have a larger *SEA*. For the EVA foams, the greater the density, the greater the mass, and the higher the cost. Considering lightweight and cost factors within the safe range, when choosing EVA foam, we can choose a less dense EVA foam material.

## 4. Conclusions

Herein, the dynamic crushing responses and mechanical characteristics of EVA foams with different densities and thicknesses under impacting were experimentally studied. The maximum contact force, maximum placement, maximum strain, absorbed energy, and *SEA* were analyzed, and the following conclusions were drawn:

The contact force–time curves of the EVA foams presented the same trend and were similar to the dynamic compression characteristics of other foams. The contact force–time curves of different thicknesses and densities show a symmetrical trend of first increasing and then decreasing to some extent. 

Within the range of impact energy and thickness studied, the specimen thickness has no effect on the energy absorption capacity, and the specimen thickness has no significant influence on the absorbed energy.

Within the range of impact energy and density studied, the density has a great influence on the energy absorption capacity. For a specimen of any density, the absorbed energy increases with the increase in impact energy, and the relationship between absorbed energy and impact energy is linear.

In this case, within a certain range of dynamic crushing conditions, there was an optimal thickness and density range for each cushioning material. In the optimization design of cushioning packaging, in order to prevent resource waste caused by excessive packaging or damage caused by insufficient packaging, the dynamic characteristics and energy absorption performance of EVA foams with different thicknesses and densities should be fully compared, and an optimal scheme should be selected according to the comparison results.

In the future, we can also conduct a more in-depth and comprehensive study on the dynamic cushioning properties of EVA foams, including the influence of the preparation method, the structure of the bubble hole, the size of the porosity, ambient temperature and humidity, etc. This study will provide theoretical support and the basis for the cushioning packaging design.

## Figures and Tables

**Figure 1 polymers-15-03016-f001:**
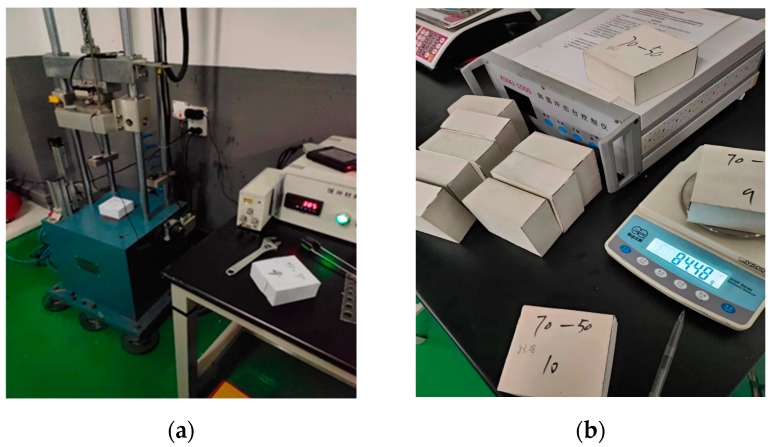
The equipment and EVA specimens: (**a**) testing equipment; (**b**) EVA specimens; (**c**) the SEM structure of the specimen with a hardness of 45 HA; (**d**) the SEM structure of the specimen with a hardness of 70 HA.

**Figure 2 polymers-15-03016-f002:**
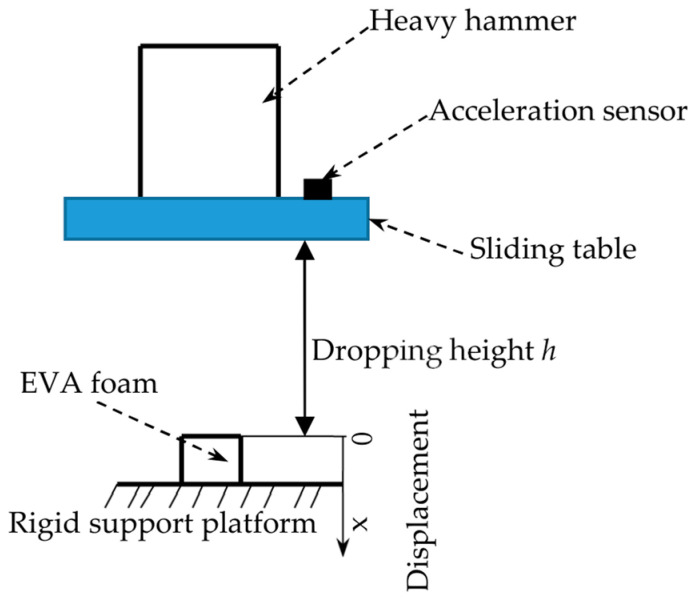
Diagram of drop impact test of heavy hammer.

**Figure 3 polymers-15-03016-f003:**
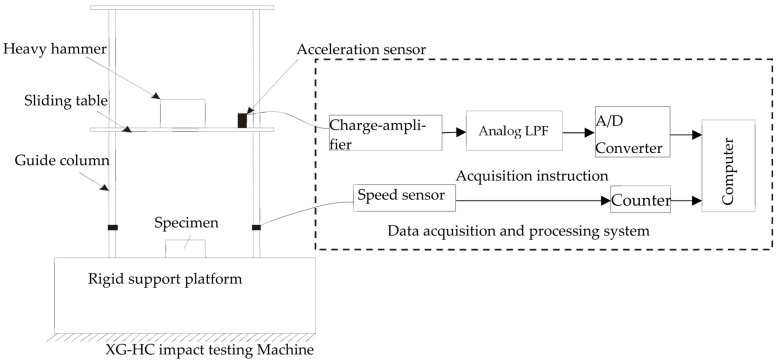
System principles of XG-HC impact testing machine.

**Figure 4 polymers-15-03016-f004:**
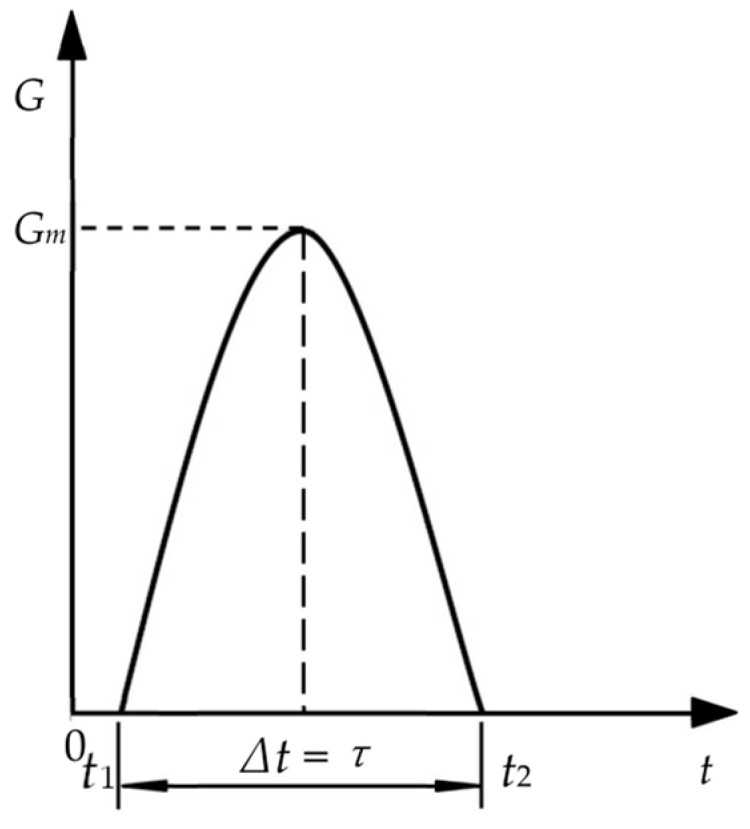
Acceleration versus time curves [26].

**Figure 5 polymers-15-03016-f005:**
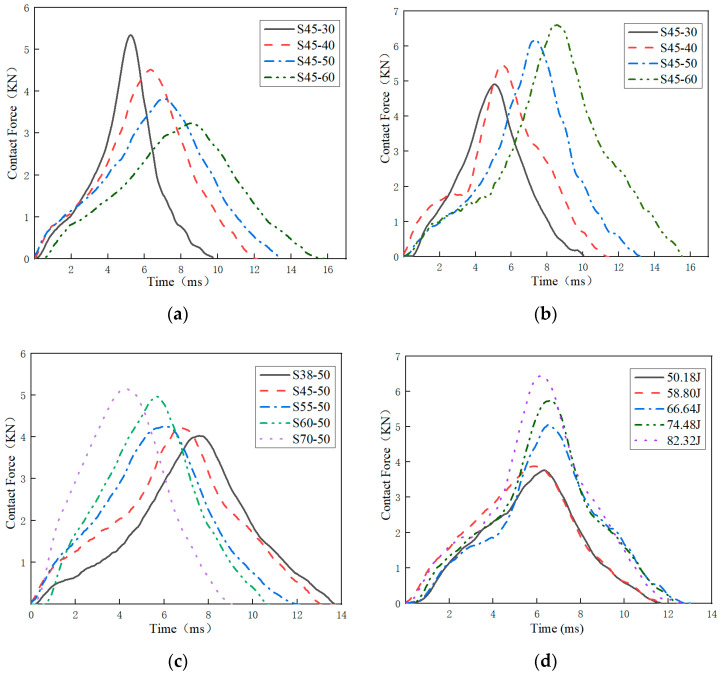
Contact force–contact time curves (**a**) at an impact energy of 66.64 J for different thickness specimens; (**b**) at an impact energy-to-thickness ratio of 1680 J/m; (**c**) at an impact energy of 66.64 J for different density specimens; (**d**) at different energies impacting on the specimen of S38-50.

**Figure 6 polymers-15-03016-f006:**
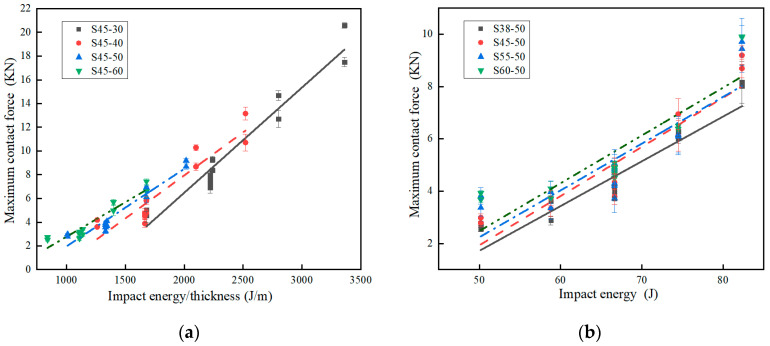
Maximum contact force curves. (**a**) Curves of maximum contact force vs. impact energy-to-thickness ratio for the different thickness specimens; (**b**) curves of maximum contact force vs. impact energy for the different densities.

**Figure 7 polymers-15-03016-f007:**
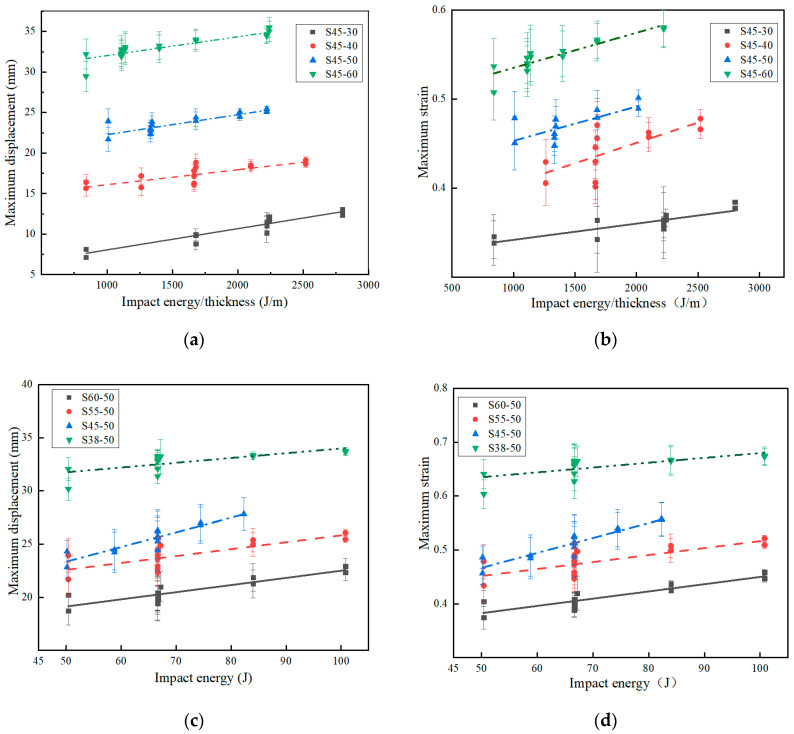
Maximum contact force, maximum displacement, and maximum strain curves of specimens with different thicknesses and different densities. (**a**) Curves of maximum displacement vs. impact energy-to-thickness ratio for the different thickness specimens; (**b**) curves of maximum displacement vs. impact energy-to-thickness ratio for the different thickness specimens; (**c**) curves of maximum displacement vs. impact energy for the different density specimens; (**d**) curves of maximum strain vs. impact energy for the different density specimens.

**Figure 8 polymers-15-03016-f008:**
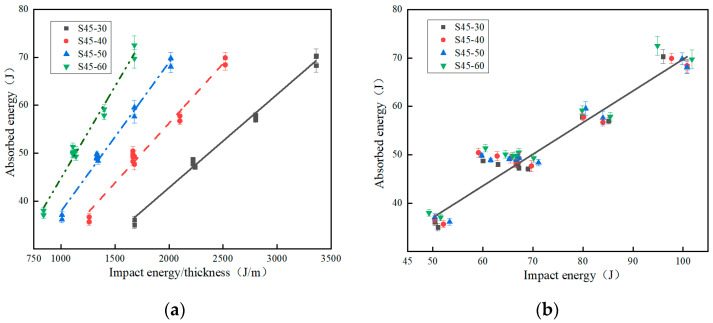
Energy absorption characteristic curves of specimens with different thicknesses and different densities. (**a**) Curves of absorbed energy versus impact energy-to-thickness ratios for the EVA specimens with different thicknesses; (**b**) curves of absorbed energy versus impact energies for the EVA specimens with different thicknesses; (**c**) curves of *SEA* versus impact energy for the EVA specimens with different thicknesses; (**d**) curves of absorbed energy versus impact energy for the EVA specimens with different densities; (**e**) curves of *SEA* versus impact energy for the EVA specimens with different densities.

**Table 1 polymers-15-03016-t001:** Experiment schemes of the EVA specimens with different thicknesses.

Specimen	Thickness (mm)	Theoretical Impact Energy (J)	Theoretical Impact Energy-to-Thickness Ratio (J/m)	Number of Specimens
S45-30	30	50.4	1680	5
		66.64	2221	5
		67.2	2240	5
		84	2800	6
		100.8	3360	6
S45-40	40	50.4	1260	5
		66.64	1666	5
		67.2	1680	5
		84	2100	6
		100.8	2320	6
S45-50	50	50.4	1008	5
		66.64	1333	5
		67.2	1344	5
		84	1680	6
		100.8	2016	6
S45-60	60	50.4	840	5
		66.64	1110	5
		67.2	1120	5
		84	1400	6
		100.8	1680	6

**Table 2 polymers-15-03016-t002:** Experiment schemes of the EVA specimens with different densities.

Specimen	Density (kg/m^3^)	Theoretical Impact Energy (J)	Theoretical Impact Energy/Thickness Ratio (J/m)	Number of Specimens
S38-50	70	50.18	1003	5
		58.80	1176	5
		66.64	1333	5
		74.48	1489	6
		83.32	1666	6
S55-50	110	50.18	1003	5
		58.80	1176	5
		66.64	1333	5
		74.48	1489	6
		83.32	1666	6
S60-50	150	50.18	1003	5
		58.80	1176	5
		66.64	1333	5
		74.48	1489	6
		83.32	1666	6
S70-50	175	50.18	1003	5
		58.80	1176	5
		66.64	1333	5
		74.48	1489	6
		83.32	1666	6

**Table 3 polymers-15-03016-t003:** Maximum contact force, maximum displacement, and maximum strain of specimens with different thicknesses under different impact energy/thickness ratios.

Specimen	Theoretical Impact Energy/Thickness Ratio (J/m)	Maximum Displacement (mm)	Maximum Strain	Maximum Contact Force (KN)
S45-30	1680 ± 2	10.92 ± 0.10	0.36 ± 0.02	4.91 ± 0.20
	2221 ± 3	12.22 ± 0.20	0.41 ± 0.02	5.63 ± 0.20
	2240 ± 3	14.16 ± 0.20	0.47 ± 0.02	10.56 ± 0.50
	2800 ± 4	15.36 ± 0.20	0.51 ± 0.01	13.79 ± 0.50
	3360 ± 4	16.01 ± 0.20	0.53 ± 0.02	17.75 ± 0.50
S45-40	1260 ± 2	15.751 ± 0.20	0.41 ± 0.02	3.71 ± 0.40
	1666 ± 2	17.83 ± 0.20	0.45 ± 0.02	4.52 ± 0.40
	1680 ± 2	18.83 ± 0.10	0.47 ± 0.02	5.46 ± 0.40
	2100 ± 3	18.50 ± 0.20	0.46 ± 0.01	7.03 ± 0.40
	2520 ± 3	19.12 ± 0.20	0.48 ± 0.02	12.63 ± 0.50
S45-50	1008 ± 2	21.69 ± 0.20	0.45 ± 0.01	2.06 ± 0.20
	1333 ± 2	23.06 ± 0.20	0.46 ± 0.02	3.81 ± 0.30
	1344 ± 2	23.87 ± 0.20	0.48 ± 0.03	4.68 ± 0.30
	1680 ± 2	24.39 ± 0.20	0.49 ± 0.02	6.30 ± 0.30
	2016 ± 3	25.07 ± 0.10	0.50 ± 0.02	8.48 ± 0.40
S45-60	840 ± 2	29.46 ± 0.24	0.51 ± 0.01	2.60 ± 0.20
	1110 ± 2	32.20 ± 0.20	0.54 ± 0.03	3.24 ± 0.20
	1120 ± 2	33.09 ± 0.20	0.55 ± 0.01	4.39 ± 0.20
	1400 ± 2	33.23 ± 0.20	0.55 ± 0.01	5.48 ± 0.30
	1680 ± 3	34.01 ± 0.30	0.57 ± 0.02	6.87 ± 0.40

**Table 4 polymers-15-03016-t004:** Maximum contact force, maximum displacement, and maximum strain of specimens with different densities under different impact energies.

Specimen	Theoretical Impact Energy (J)	Maximum Displacement (mm)	Maximum Strain	Maximum Contact Force (KN)
S60-50	50.18 ± 1	18.72 ± 0.30	0.37 ± 0.01	3.79 ± 0.20
	58.80 ± 1	19.475 ± 0.40	0.39 ± 0.01	3.91 ± 0.20
	66.64 ± 2	20.41 ± 0.30	0.41 ± 0.01	4.73 ± 0.30
	74.48 ± 2	21.27 ± 0.40	0.43 ± 0.01	6.44 ± 0.40
	82.32 ± 3	22.91 ± 0.50	0.46 ± 0.01	9.90 ± 0.50
S55-50	50.18 ± 1	21.89 ± 0.30	0.43 ± 0.01	3.59 ± 0.20
	58.80 ± 1	22.46 ± 0.30	0.45 ± 0.01	3.66 ± 0.20
	66.64 ± 2	23.85 ± 0.40	0.49 ± 0.01	4.49 ± 0.30
	74.48 ± 2	24.76 ± 0.50	0.49 ± 0.01	6.12 ± 0.40
	82.32 ± 3	26.07 ± 0.50	0.52 ± 0.02	9.58 ± 0.50
S45-50	50.18 ± 1	22.85 ± 0.30	0.46 ± 0.01	2.88 ± 0.20
	58.80 ± 1	24.49 ± 0.30	0.49 ± 0.01	3.53 ± 0.30
	66.64 ± 2	26.31 ± 0.40	0.53 ± 0.02	4.57 ± 0.40
	74.48 ± 2	26.78 ± 0.50	0.54 ± 0.02	6.53 ± 0.40
	82.32 ± 3	27.83 ± 0.50	0.56 ± 0.02	8.94 ± 0.50
S38-50	50.18 ± 1	30.16 ± 0.30	0.60 ± 0.03	2.66 ± 0.20
	58.80 ± 1	32.08 ± 0.30	0.64 ± 0.03	3.25 ± 0.20
	66.64 ± 2	33.16 ± 0.40	0.66 ± 0.03	3.94 ± 0.30
	74.48 ± 2	33.24 ± 0.50	0.66 ± 0.03	6.15 ± 0.40
	82.32 ± 2	33.76 ± 0.50	0.67 ± 0.03	8.09 ± 0.50

**Table 5 polymers-15-03016-t005:** Absorbed energy for the EVA specimens with different thicknesses under different impact energies.

Specimen	Impact Energy (J)	Impact Energy/Thickness (J/m)	Absorbed Energy (J)	*SEA* (J/g)
S45-30	50.40 ± 1	1680 ± 3	36.11 ± 1	2.81 ± 0.10
	66.64 ± 2	2221 ± 4	47.77 ± 1	2.81 ± 0.10
	67.20 ± 2	2240 ± 4	47.26 ± 1	2.76 ± 0.10
	84.00 ± 3	2800 ± 4	57.01 ± 2	2.66 ± 0.10
	100.80 ± 3	3360 ± 5	68.29 ± 3	2.66 ± 0.10
S45-40	50.40 ± 1	1260 ± 3	36.69 ± 1	2.85 ± 0.10
	66.64 ± 2	1666 ± 3	48.27 ± 2	2.84 ± 0.10
	67.20 ± 2	1680 ± 3	49.32 ± 2	2.88 ± 0.10
	84.00 ± 3	2100 ± 4	57.78 ± 3	2.70 ± 0.10
	100.80 ± 3	2520 ± 4	69.92 ± 3	2.72 ± 0.10
S45-50	50.40 ± 1	1008 ± 3	37.12 ± 1	2.89 ± 0.10
	66.64 ± 2	1333 ± 3	48.89 ± 2	2.88 ± 0.10
	67.20 ± 2	1344 ± 3	49.14 ± 2	2.89 ± 0.10
	84.00 ± 3	1680 ± 3	57.62 ± 3	2.69 ± 0.10
	100.80 ± 3	2016 ± 4	69.83 ± 3	2.72 ± 0.10
S45-60	50.40 ± 1	840 ± 2	37.04 ± 2	2.88 ± 0.10
	66.64 ± 2	1110 ± 2	49.74 ± 3	2.93 ± 0.10
	67.20 ± 2	1120 ± 3	50.51 ± 3	2.95 ± 0.10
	84.00 ± 3	1400 ± 3	59.16 ± 3	2.76 ± 0.10
	100.80 ± 3	1680 ± 3	72.52 ± 4	2.82 ± 0.10

**Table 6 polymers-15-03016-t006:** Absorbed energy for the EVA specimens with different densities under different impact energies.

Specimen	Impact Energy (J)	Absorbed Energy (J)	*SEA* (J/g)
S38-50	50.18 ± 1	36.84 ± 1	1.05 ± 0.10
	58.80 ± 1	40.81 ± 2	1.17 ± 0.10
	66.64 ± 2	47.77 ± 2	1.36 ± 0.10
	74.48 ± 2	51.64 ± 3	1.48 ± 0.10
	82.32 ± 3	53.58 ± 3	1.53 ± 0.10
S45-50	50.18 ± 1	36.90 ± 1	0.88 ± 0.05
	58.80 ± 1	42.15 ± 1	1.00 ± 0.10
	66.64 ± 2	48.27 ± 1	1.15 ± 0.10
	74.48 ± 2	50.04 ± 2	1.19 ± 0.10
	82.32 ± 3	51.35 ± 2	1.22 ± 0.10
S55-50	50.18 ± 1	36.78 ± 1	0.67 ± 0.02
	58.80 ± 1	44.53 ± 1	0.81 ± 0.02
	66.64 ± 2	48.52 ± 1	0.88 ± 0.02
	74.48 ± 3	51.99 ± 2	0.99 ± 0.02
	82.32 ± 4	53.90 ± 2	1.11 ± 0.1
S60-50	50.18 ± 1	38.55 ± 1	0.51 ± 0.01
	58.80 ± 1	43.42 ± 1	0.58 ± 0.01
	66.64 ± 2	49.74 ± 2	0.66 ± 0.01
	74.48 ± 3	52.72 ± 2	0.70 ± 0.01
	82.32 ± 4	55.84 ± 2	0.74 ± 0.01

## Data Availability

Not applicable.

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
