# Peer review of "Dynamic Crushing Behavior of Ethylene Vinyl Acetate Copolymer Foam Based on Energy Method"

_polymers, 2023, doi:10.3390/polym15143016_

Round 1

Reviewer 1 Report

The purpose of this study was to experimentally elucidate the dynamic crushing mechanism and performance of ethylene vinyl acetate copolymer foams, and the effects of density and thickness on the mechanical behavior and energy absorption properties were analyzed under dynamic impact loading. The subject is interesting and the manuscript has potentials. However, some modifications are required before the final decision.

1. Line 12, please avoid missing spaces between values and units, 66.64J. Each value should have a space with its unit. Please address this issue throughout the manuscript. For instance, lines 15, 18, 21, 104, 114, and etc.

2. SEA acronym has been used in line 13; however, its full form has been introduced in line 17. Also, its full form has been repeated in lines 22, 28, and 30. Each acronym should be introduced with its full form only in its first mentioning. Afterwards, its acronym should be used.

3. EPP acronym has been used in line 78; however, its full form has not been introduced.

4. What is the method of measurement of hardness of the EVA foam? What is 38 A? For instance, it may be Rockwell A.

5. Citation of equations in section 2.4 is really weak. All of the equations should be cited with supporting referencing where they are required. For instance, Equations (1), (2), (3), (4), and etc.

6. The captions of Tables 1 and 2 show that the results are experimental; however, the results seem to be theoretical. Please clarify this.

7. The experimental results should be presented with corresponding standard deviations.

8. The authors claimed that the samples are EVA foams. However, there is no evidence to support this claim, such as SEM micrographs.

9. Future directions and insights should be addressed in a separate paragraph.

Author Response

Dear reviewer,

     We are very pleased to learn from your letter about the revision of my manuscript (Dynamic crushing behavior of ethylene vinyl acetate copolymer foam based on energy method. Manuscript ID: polymers-2479319). Thanks for your attention and your helpful comments and advice. I have revised the manuscript according to the comments from the reviewers. Please see the attachment.

Reviewer 2 Report

The manuscript entitled “Dynamic crushing behavior of ethylene vinyl acetate copolymer foam based on energy method” describes an investigation into the dynamic crushing mechanism exhibited by ethylene vinyl acetate copolymer (EVA) foam as well as its performance and how it affects its mechanical behavior as well as its energy absorption when it is subjected to various mechanical loadings.

            Overall, the research appears to have been well-executed and this work is of interest both from a fundamental scientific as well as a practical standpoint. The manuscript is generally well-written. With regard to the numbers, in some cases the numbers and the corresponding units are not separated by a space; it may be necessary to add a space in those cases (with the exception of the unit “%”, which does not require a space between it and the corresponding number). Overall, I believe that this manuscript is suitable for publication pending minor revisions, such as those which are outlined below.

Line 14, 20 and 23: Error margins may be needed for increases in maximum contact force, maximum displacement, maximum strain, absorbed energy and specific energy displacement (SEA) .

Line 19: “Stain” should be changed to “strain”.

Line 39: The term “children floor” is unclear.

Line 76: “However, few information about” can be changed to “However, there a paucity of information about” or “There is a lack of information about”.

Line 104: For some of the numbers listed here, there does not seem to be a space after the comma.

Line 117: “is 7 kg and 50 kg respectively.” Can be changed to “is 7 kg and 50 kg, respectively.”.

Line 133: A period will be needed at the end of the caption for Figure 1.

Line 168: A period will be needed at the end of the caption for Figure 3.

Lines 231 and 235: “showed” can be changed to “shown”.

Lines 236-237, Table 1, top of second column from the left: “Thichness” should be changed to “Thickness”.

Lines 254-255, 263, and 309: “30mm to 60 mm” can be changed to “30 mm to 60 mm” or “30 to 60 mm”. Note also that a space should be added between a number and the corresponding unit (30 mm instead of 30mm). This change may be needed for other numbers throughout the manuscript as well.

Line 273: “(13.892ms–10.568ms).As shown” should be changed to “(13.892 ms – 10.568 ms). As shown” or “(13.892 – 10.568 ms). As shown”.

Line 288: “showed” can be changed to “shown”.

Line 320: “and 0.6632-0.4082) respectively.” Can be changed to “and 0.6632-0.4082), respectively.”

Lines 324-325: Error margins may be needed for some of the values reported in Table 3.

Lines 326-327: Error margins may be needed for some of the values reported in Table 4.

Line 341, Figure 6 caption: “specimens(c)” can be changed to “specimens (c)”.

Line 342, Figure 6: “specimens(d) can be changed to “specimens (d)”.

Lines 369 and 375: “30mm to 60mm” can be changed to “30 mm to 60 mm” or “30 to 60 mm”.

Line 377: “74.6%)respectively.” Can be changed to “74.6%), respectively.”.

Line 395: “the absorbed energy of specimen with” can be changed to “the absorbed energy of a specimen with”.

Lines 414-416, Table 5: Error margins may be needed for some of the values reported in Table 5.

Lines 418-419, Table 6: Error margins may be needed for some of the values reported in Table 6.

Overall, the manuscript is well-written. I have provided some minor suggestions in the general comments for some minor revisions. 

Author Response

(The authors gave the same response as above.)

Reviewer 3 Report

The manuscript “Dynamic crushing behavior of ethylene vinyl acetate copolymer foam based on energy method” by Yueqing Xing et al. presents results of the experimental study of ethylene vinyl acetate foams under dynamic loadings using the dropping heavy hummer technique. The methodology applied and results obtained could be of interest for the community of “Polymers” readers specialized in the field of polymer mechanics. The manuscript fits the topics of the journal. However, many questions rise up during manuscript reading, which should be answered before the manuscript acceptance.

1. It seems useful to specify the studied samples of EVA foams not only by the hardness and density, but also the porosity and average pore size. The latter parameters are also crucial for the foam mechanical properties.

2. After dynamical impact, the foam samples should partially relax to their initial states (in shapes and dimensions). The discussion of this point supported by some quantitative data is very desirable.

3.    The weak point of this work is the practical absence of any statistical analysis of the obtained data and estimates of measurement errors. All data are given throughout the text using 4 or 5 significant digits, with no confidence intervals. Moreover, the displacement values are given with the accuracy of one micrometer, and the energies are defined with the accuracy of 1 mJ (see Tables 3-6). At the same time, the graphs show a rather significant scatter of the obtained data. Detailed analysis of measurement errors and corresponding correction of the presented data are necessary.

4. The authors use the sine approximation (Eq. 3) for the time-dependent deceleration of the heavy hummer with sliding table. At the same time, the obtained experimental curves for the impact force (Fig. 4) exhibit strong deviation from the sine law. The reasons for such deviation should be discussed and its influence on the measurement accuracy should be analyzed.

5. Regarding the “dynamic stress” (Eq. 15, line 214). Does the physical dimension of this parameter differ from the static stress” dimension (N/m^2). Following Eq. 15, this dimension should be N/mS^2. What is the reason for its introduction? As I guess, it is not used for data analysis and interpretation. Please, provide the physical meaning and necessity of its application.

6. The author use the term “dynamic energy” (line 211) with the dimension of pressure or energy density (N/m^2). This requires more detailed explanations.

Moderate editing of English language required.

Author Response

(The authors gave the same response as above.)

Round 2

Reviewer 1 Report

The revision is satisfactory. The manuscript is recommended for publication in its current format.

Reviewer 3 Report

The authors have thoroughly revised the manuscript; as a result, its quality has improved compared to the previous version. Of course, the question of the influence of the foam structure on its dynamic response is still open. Nevertheless, the current version of the manuscript is acceptable for publication.

Minor editing of English language is required.